# Removal of Methylene Blue from Water Using Magnetic GTL-Derived Biosolids: Study of Adsorption Isotherms and Kinetic Models

**DOI:** 10.3390/molecules28031511

**Published:** 2023-02-03

**Authors:** Shifa Zuhara, Snigdhendubala Pradhan, Yahya Zakaria, Akshath Raghu Shetty, Gordon McKay

**Affiliations:** 1Division of Sustainable Development, College of Science and Engineering, Hamad Bin Khalifa University, Education City, Qatar Foundation, Doha P.O. Box 5825, Qatar; 2Qatar Environment and Energy Research Institute, Hamad Bin Khalifa University, Qatar Foundation, Doha P.O. Box 34110, Qatar

**Keywords:** biosolid, activation, dye removal, isotherm, kinetics, thermodynamics

## Abstract

Global waste production is significantly rising with the increase in population. Efforts are being made to utilize waste in meaningful ways and increase its economic value. This research makes one such effort by utilizing gas-to-liquid (GTL)-derived biosolids, a significant waste produced from the wastewater treatment process. To understand the surface properties, the biosolid waste (BS) that is activated directly using potassium carbonate, labelled as KBS, has been characterized using scanning electron microscopy and energy dispersive X-ray spectroscopy (SEM-EDS), X-ray photoelectron spectroscopy (XPS), X-ray powder diffraction (XRD), and Brunauer–Emmett–Teller (BET). The characterization shows that the surface area of BS increased from 0.010 to 156 m^2^/g upon activation. The EDS and XPS results show an increase in the metal content after activation (especially iron); additionally, XRD revealed the presence of magnetite and potassium iron oxide upon activation. Furthermore, the magnetic field was recorded to be 0.1 mT using a tesla meter. The magnetic properties present in the activated carbon show potential for pollutant removal. Adsorption studies of methylene blue using KBS show a maximum adsorption capacity of 59.27 mg/g; the adsorption process is rapid and reaches equilibrium after 9 h. Modelling using seven different isotherm and kinetic models reveals the best fit for the Langmuir-Freundlich and Diffusion-chemisorptionmodels, respectively. Additional thermodynamic calculations conclude the adsorption system to be exothermic, spontaneous, and favoring physisorption.

## 1. Introduction

The global population is ever-growing, recently surpassing 8 billion people, and wastewater production is also on the rise and projected to increase by 51% by the year 2030 from the 2020 levels of 380 billion m^3^ [1]. Water pollution associated with high amounts of aromatic pollutants with dark colors and weakly biodegradable, complex components generated from the dyeing industry is known to be problematic [2]. For the longest time, the wastewater produced from the dyeing industry was considered the ‘hot spot’ of wastewater treatment research. The common treatment methods included chemical oxidation, membrane treatment, photo-degradation, biological treatment, micellar solubilization, adsorption, and others [3,4,5]. Adsorption, the process by which the substances present in liquids or gases adhere onto the surfaces of solid adsorbents [6], is generally applied for water purification to remove impurities. It is known to be one of the most preferable treatment technologies for dye removal as it is inexpensive, simple, and highly efficient [7]. The commercial production of activated carbon (AC) is expensive; therefore, ongoing research on biomass and wastes for adsorption is being carried out extensively.

Population growth also leads to an exponential rise in the waste produced, causing ecological and social burdens along with environmental ramifications. The management techniques are mostly based on incineration, open dumping, and landfilling, which are associated with environmental risks because of the associated greenhouse gas emissions [8]. Resource utilization or producing value-added products from waste and biomass is being widely studied to reduce the total amount of waste produced [9,10,11]. Among several kinds of waste, the inevitable wastes produced from biological wastewater treatment plants (biosolids/sewage sludge) are significant [12]. The waste is upgraded by activation through physical or thermal (air, carbon dioxide, or steam) and chemical (acid, alkali, neutral) means. The process is known to improve the surface properties of the feed in a single-stage activation system and biochar in a two-stage activation system [13]. In addition to the surface properties, certain elements and bonds are also known to improve the adsorption process. Generally, sludge samples are known to have high iron content based on the treatment method and chemicals used during treatment [14]. Research has branched out to adding nano-valent iron to the sludge-based biochar for increased magnetism [15,16]. There is also increased interest in magnetic iron oxide nanoparticles (MNPs) as an adsorbent and catalyst for pyrolysis studies [17].

For this study, AC is produced from GTL-derived treated sewage sludge (biosolids). Around 6000 tons of biosolids are generated annually in the world’s largest GTL plant in Qatar [18]. Since sludge/biosolid-derived AC is rich in functional groups with improved surface area and pore volume, it can be considered beneficial for removing pollutants from water [19]. The targeted pollutant for this work is methylene blue (MB): an aromatic organic compound and cationic dye known to be highly carcinogenic in nature [19] and frequently regarded as a standard for adsorbent testing. The objectives of this study are the following: Initially, to activate GTL-derived biosolids (BS) using potassium carbonate (KBS). Then, the biosolid pristine and activated samples were characterized in order to understand the effects on the contents and composition of the samples. Furthermore, the effects of varying concentrations, contact time, pH, and temperature on the adsorption of MB using KBS were investigated. Using the generated adsorption data, isotherm and kinetical modelling was carried out to evaluate best-fit models. Finally, thermodynamic calculations from the isotherm data are performed to understand the adsorption system better for future applications and equilibrium reaction boundary limits.

## 2. Results and Discussion

### 2.1. Characterization of Samples

The impregnation ratios 1:1 and 1:2 (sample:activating ratios) generated ACs with low surface area and porosity, which seems unfit to be used for water treatment (Appendix A). Table 1 reveals the common characteristics of pristine biosolid (BS) and activated BS samples (KBS) prepared by 1:3 (sample:activating agent) ratio. The biosolid (pristine and activated) samples are alkaline and negatively charged. The yield calculated for KBS is 33.25%, which is better than the reported literature on single-stage ACs from biosolids [20]. There is a significant improvement in the surface area, increasing from 0.010 to 156.56 m^2^/g, and pore volume, from 0.021 to 0.235 cm^3^/g, upon activation (Appendix A), better than some reported studies [21].

The SEM images reveal the lack of porosity and surface area in the pristine raw material sample and show increased fragmentation and surface area in the activated sample (Figure 1). Elemental microanalysis (Table 2) in the pristine biosolid sample shows a high content of metals such as iron, calcium, magnesium, and potassium—the natural presence of these metals improves pore development due to the sintering process [22].

Further EDS analysis of the sample before and after activation, shown in Table 2, reveals that an increase in temperature concentrates elements, mainly phosphorous, calcium, and iron, present in the sample. The iron content in terms of mass percentage increased from 6.53% to 31%, showing a good improvement in magnetic properties; this is also evident from the brown color of the sample after activation (Appendix A). Due to the presence of the oxygen-containing activating agent potassium carbonate, there is an increase in the potassium and oxygen recorded upon activation. Since it is a relative percentage, the carbon and nitrogen seem to have reduced with activation. XPS results, shown in Table 3 and Figure 2, further confirm the reduction of carbon, nitrogen, and sulfur. Furthermore, C1s-related chemical state analysis shows the presence of the peaks C-C/C-H at 284.8 eV, C-O/C-N at 286.2 eV, C=O at 287.8 eV, and -CO3/O-C=O at 289.2 eV for the BS sample. While for KBS, C1s-related chemical state analysis shows the presence of the peaks C-C/C-H at 284.8 eV, C-O/C-N at 286.3 eV, C=O at 287.7 eV, and -CO3/O-C=O at 289.3 eV, as well as K2p3/2 and K2p1/2 at 292.6 eV and 295.4 eV, respectively, as shown in Figure 2b.

The deconvolution method is detailed in previous reports [23,24,25]. The XPS results show the presence of C mainly as C-C/C-H for the BS sample with a smaller presence of C-O/C-N and with a minor presence of C=O and -CO3/O-C=O. However, KBS shows the presence of mainly C-C/C-H as well as a smaller presence of C-O and a minor presence of C=O. After the activation, the -CO3/O-C=O peak relative area has increased compared to C-C/C-H, which is related to the presence of carbonates (-CO3) at the surface. This activation has also kept some K at the surface. The increased presence of metals and specific carbon bonds may prove beneficial for pollutant removal studies [13].

The XRD analysis of KBS shows the prominent presence of magnetite and potassium iron oxide (K_2_Fe_4_O_7_, KFeO_2_), while calcite was seen to be the only ‘major’ deposit detected on the pristine sample (Figure 3). The diffraction peaks at 35.2° matched well with the PDF spectrum of Fe_3_O_4,_ showing the presence of magnetic properties after activation [26]. Further analysis of KBS using a tesla meter revealed a magnetic field of 0.1 mT, confirming the sample is weakly magnetic.

### 2.2. Adsorption Experiments

The adsorption experiments using KBS revealed that the maximum adsorption capacity of methylene blue is 59.27 mg/g at 40 °C and pH 7 (Figure 4). The effect of pH is seen to be significant as the adsorption capacity increased from 9.89 mg/g at pH 2 to reach the maximum at pH 7, with little change in further alkaline conditions (Appendix A). This could be due to increased electrostatic interaction between negatively charged KBS and cationic MB at higher pH due to the lack of competition by H+ ions in water for vacant sites on the AC. The adsorption capacities are comparable to some waste adsorbents reported in the literature (Table 4). More recently, potassium-carbonate-activated samples have been shown to have better properties compared to other alkali methods. One such study reported to have a better yield, surface area, and pore volume compared to potassium hydroxide, thus enabling a higher adsorption capacity for larger molecules such as methylene blue [27].

### 2.3. Isotherm Modelling

The adsorption isotherm shows the distribution of adsorbed molecules between MB and KBS at an equilibrium state. The adsorption data were fitted to seven different models described in Appendix A. The two best-fit models, based on the least sum of squares error (SSE), are the Langmuir–Freundlich (LF) model followed by the Langmuir model (Table 5, Figure 5). The LF model is a modification of the Langmuir and Freundlich models used to predict a heterogenous system, also circumventing the limitations related to increased adsorbate concentrations of the Freundlich model. Based on the assumptions of the LF model (Appendix A), the adsorption system localizes without adsorbate–adsorbate interaction. Additionally, the second best-fit model the Langmuir model suggests the adsorption took place on a monolayer which is also an assumption of the LF model. The results from this section prove that there is a high availability of vacant sites on the surface for adsorption.

### 2.4. Contact Time Study and Kinetic Modelling

This study also concludes that contact time plays an essential part in MB adsorption. The influence of time from 0 to 1440 min (12 h) is recorded and plotted in Figure 6a. Most of the adsorption process took place rapidly in the first 2 h with high removal efficiencies. Furthermore, when 500 ppm MB concentration was used for adsorption, the adsorption reached a plateau after 9 h, reaching a removal efficiency of 11.8%. Kinetic modelling of the adsorption data using the contact time data (Co = 500 ppm) using the seven models described in Appendix A revealed that the diffusion–chemisorption, pseudo-first order (PFO),and pseudo-second order (PSO) models have the better fits based on the least SSE error values (Table 6, Figure 6b). The model that correlates best with the data is the Diffusion-chemisorption (DF) model. The DF model reveals that much like the isotherm modelling results, the adsorption system favors the sorption of adsorbate onto a heterogeneous surface and that the rate of change of concentration in solid phase correlates to the rate of mass of transfer of pollutant in fluid phase during adsorption (Appendix A).

### 2.5. Effect of Temperature and Thermodynamic Calculations

The isotherm curves of the sample were plotted by data obtained at 20, 30, and 40 °C (Figure 7a). The results show that adsorption trends were the same for all conditions. The data from the final point at 500 ppm (Ce and qe) for all three temperatures were used for the calculation of ∆G°. Figure 7b shows the data plot of KBS (1/T Vs. lnK_d_); the plot showed excellent correlation (R^2^ > 0.998); it is evident the ∆H °value is negative, and ∆S ° is positive (Table 7). The negative ∆H ° value shows that the reaction is exothermic, and the positive ∆S ° reveals the randomness of the solid-liquid interface of the adsorption system, and a good attraction is observed between MB and the ACs.

The isotherm curves of the sample were plotted by data obtained at 293.15, 303.15, and 313.15 K (Figure 7a). The results show that the adsorption trends were the same for all conditions. The data from the final point at 500 ppm (Ce and qe) for all three temperatures were used for the calculation of ∆G°. Figure 7b shows the data plot of KBS (1/T vs. lnK_d_) and the detailed calculations are shown in Appendix A; the plot showed excellent correlations (R^2^ > 0.998) and it is evident the ∆H °values are negative, and ∆S ° is positive (Table 7). The negative ∆H ° values show that the reaction is exothermic, and the positive ∆S ° reveals the randomness of the solid–liquid interface of the adsorption system, and a good attraction is observed between MB and the ACs. Furthermore, the ∆G ° calculated is negative, showing that the adsorption system was spontaneous, suggesting a physisorption process (as the value is in between −20 KJ/mol and 0 KJ/mol) [34]. This is also supported by the ∆H value showing lesser than 80 KJ/mol, indicating a physisorption system (Table 7). There are several such adsorption systems reported in the literature [35,36]. The slight ∆G ° decrease with an increase in temperature shows that this MB adsorption study favors higher temperatures. Furthermore, the activation energy Ea measurement from the slope between lnK_d_ and 1/T shows a value of 22.7 KJ/mol, indicating the reaction is quick as it is below 40 KJ/mol [37].

The negative surface charge of the samples (Table 1) attracts the positively charged cationic colored dye ion of methylene blue. Given the experimental conditions, it is possible that electrostatic interactions played a key role in the adsorption mechanism—this is supported by another study on a K_2_CO_3_-activated grass waste sample used for MB removal [38].

## 3. Materials and Methods

### 3.1. Activation

The biosolid sample was obtained from the Pearl Shell GTL plant in Qatar in the dried form. Potassium carbonate (1M) was added to the sample at an impregnation ratio of 1:1, 1:2, and 1:3 (sample:activating agent) and mixed at 100 rpm for 24 h in an auto shaker. The samples were then dried at 105 °C for 8 h. The powdered samples were then activated in a muffled furnace at a heating rate of 10 °C/min until 700 °C for 2 h with a constant flow of nitrogen. The treated samples were washed, and the pH adjusted to neutral using 1M hydrochloric acid. The samples were dried before further use. The sample was activated using a 1:3 (sample:activating agent) ratio, labelled KBS, and was used for characterization and further adsorption experiments.

The yield was calculated as follows (Equation (1)):Yield (%) = Weight of biochar (g) /Weight of oven-dried wastes (g) × 100(1)

### 3.2. Characterization

#### 3.2.1. pH

The pH of the samples was determined using a modified ASTM standard method D3838-99 (ASTM, 217 2005). About 1 g of dried sample was put into a beaker containing 10 mL of boiling de-ionized water. The solution mixture was heated and allowed to boil for about 15 min in a sealed tube. Then, the solution was filtered employing a pre-moistened filter paper (Whatman No. 2, 110 mm diameter). 

#### 3.2.2. Zeta Potential

A zeta potential analyzer (Zetasizer Nano-ZS, Malvern P analytical, Malvern, UK) was used to analyze the charge of the char. Before using the instrument, 0.1 g of the char sample was added to 200 mL of distilled water to make a suspension of 0.5 ppm; further shaking at 150 rpm was performed for 12 h to ensure adequate mixing.

#### 3.2.3. BET

The surface area was characterized by BET (Brunauer–Emmett–Teller) nitrogen sorption at 77 K temperature with a relative pressure between 0.05 to 0.35 using a Nova 2200e surface area analyzer (Tristar3200, Micromeritics, Norcross, GA, USA). The degassing was carried out at 105 °C for 24 h. The pore volume was also analyzed by BET and estimated by the liquid adsorbate volume of nitrogen at a relative pressure of 0.99.

#### 3.2.4. SEM-EDS

Scanning electron microscopy (SEM) and energy dispersive X-ray spectroscopy (EDS) were carried out to understand the morphology and the elemental microanalysis of the surface of the samples. The powdered samples were sprinkled onto adhesive carbon tape, and any excess was blown away using compressed air. The sample was then coated with 5nm of gold (using Quoram Q150 sputter, East Sussex, UK) to make it electrically conductive for SEM analysis.

Imaging was performed at 5 KV using an ETD secondary electron detector and a Quanta650FEG FEI SEM (Hillsboro, OR, USA). The elemental microanalysis was performed at 15 KV using a Bruker Quantax EDS detector (Billerica, MA, USA). Gold was deconvoluted to zero to obtain a semi-quantitative result.

#### 3.2.5. XPS

Both pristine and activated BS samples (KBS) were characterized using surface analysis technique, namely, X-ray photoelectron spectroscopy (XPS) (ESCALAB250Xi, Thermo Fisher Scientific, East Grinstead, UK). Pass energy for high resolution scans is 20 eV and for survey scans is 100 eV. XPS was calibrated using triple high purity standards of Au, Ag, and Cu. All samples were referenced using C1s (C-C/C-H) at 284.8 eV.

#### 3.2.6. XRD

For this study, a Bruker D8 Advance X-ray diffractometer with Cu Kα radiation (ʎ = 1.5418 Å) was used at instrument settings of 40 kV and 40 mA. The scan range was set from 3° to 90°. The step size was set at 0.020°.

#### 3.2.7. Tesla Meter

The magnetic field of the sample is measured using a tesla meter (PHYWE Systeme GmbH & Co. KG, Göttingen, Germany). After zeroing the instrument of the earth’s magnetic field, the sample is measured using a hall probe under constant alternating current, setting the adjustable meter range at 0–20 mT with an accuracy of 0.01 mT.

### 3.3. Adsorption Experiments

To study the effect of varying concentrations, the adsorption studies were carried out using initial concentrations of 20, 40, 60, 80, 100, 150, 200, 250, 300, 400, and 500 ppm of MB. About 0.1 g of AC (1:3 sample:activating agent) is added to 0.1 L of MB solution in each case and stirred for 24 h at 200 rpm at 40 °C. For the temperature effect, adsorption was carried out at 20, 30, and 40 °C under the isotherm conditions mentioned previously. Similarly, to understand the effect of pH, studies were conducted at 40 °C at pH 2, 4, 6, 7, and 10. For the contact time studies, the adsorption study using 500 ppm MB solution is studied; here, samples were collected at various intervals between 0 and 1440 min.

The samples’ initial and final MB concentrations were measured using UV-VIS spectrophotometry (Shimadzu UV-3600 Plus spectrophotometer, Kyoto, Japan) by measuring the absorbance spectra at steady-state conditions using the calibration curve shown in Appendix A. After adsorption, all solutions were filtered, and then the supernatant concentration was measured. The maximum methylene blue adsorption is observed at a wavelength of 664 nm (λmax) as per previous publication [39] (refer Appendix A); therefore, all measurements for this study were taken at this wavelength. The adsorption capacity (q_e_) is calculated using Equation (2):(2)$$q_{e}=\frac{V}{m}\left(C_{0}-C_{e}\right)$$
where C_o_ is the initial concentration in ppm; C_e_ is the final equilibrium concentration in ppm; V is the solution volume in L; and m is the mass of the KBS in g.

### 3.4. Adsorption Isotherms and Kinetics

To better understand the adsorption data obtained from different initial MB concentrations, equilibrium adsorption isotherm modelling is conducted. This helps to better understand the adsorption mechanism. There are seven isotherm model studies in this important paper: Langmuir, Freundlich, Redlich–Peterson, Langmuir–Freundlich, Toth, Temkin, and Dublin–Radushkevich—the equations and descriptions are shown in Appendix A. To understand pollutant uptake rate, kinetic modelling using seven kinetic models were studied—the models are described in Appendix A. The best-fit models in both cases are selected based on the least SSE (sum of squares of errors) method.

### 3.5. Thermodynamic Calculations

Adsorption isotherms at temperatures 20 °C (293.15 K), 30 °C (303.15K), and 40 °C (313.15 K) were analyzed for thermodynamic properties previously described in the literature [34,40,41]. The main focus is to study the enthalpy, entropy, and Gibbs free energy, which gives information on spontaneity. The equations used to calculate Gibbs free energy (Equations (3) and (4)):(3)$$∆G{}^{\circ}=-\mathrm{RT}\ln K_{d}$$
(4)∆G°=∆H°−T∆S

R = universal constant of 8.314 J/mol.K; T = temperature in Kelvin; q_e_ = adsorbed pollutant amount at equilibrium in mg/g; C_e_ = left over pollutant concentration in mg/L; and K_d_ (distribution coefficient of adsorption) = $\frac{\mathrm{qe}}{\mathrm{Ce}}\;x\;\frac{m}{V}$.

Additionally, the Van’t Hoff equation, evaluating the relationship between enthalpy and entropy, shows the relationship between K*_d_*, enthalpy, and entropy:ln(K_d_) = ∆S°/R = ∆H°/RT(5)

The values of ∆S° and ∆H° were calculated from the slope and intercept of the plot containing ln(K_d_) versus (1/T). In this case, the isotherm data at initial pollutant concentration of 500 ppm is used for the study. Further calculations on activation energy (E_a_) were derived from the plot ln(K_d_) versus (1/T):ln (K_Ne_) = −E_a_/RT + lnA(6)

K_d_ = thermodynamic equilibrium constant (q_e_/C_e_); R = universal constant of 8.314 J/mol.K; T = temperature in Kelvin; E_a_ = activation energy (J/mol); and A = pre-exponential factor.

## 4. Conclusions

In an attempt to convert wastes to a value-added product, this study involved utilizing gas-to-liquid (GTL)-derived biosolids, a major waste produced from the wastewater treatment process from the Qatar-Shell GTL plant. The biosolid waste was activated directly using potassium carbonate in 1:3 (sample:activating agent) ratio (KBS), and the surface area increased from 0.010 to 156 m^2^/g and pore volume from 0.021 to 0.235 cm^3^/g, which was confirmed by observing SEM scanning images. Further analysis using EDS and XPS showed an increase in metals after activation, especially iron. Deconvolution of the XPS results shows the increased presence of -CO3/O-C=O after activation of the sample. XRD analysis revealed the presence of magnetite and potassium iron oxide upon activation, and tesla meter measurement confirms magnetic properties in the AC.

The methylene blue adsorption is rapid and reaches equilibrium after 9 h, showing a maximum adsorption capacity of 59.27 mg/g, comparable to other waste adsorbents reported in the literature. Modelling using seven different isotherm and kinetic models revealed the best fit to be the Langmuir and PFO, respectively. Adsorption at different temperatures shows that the increase in temperature is favorable. It was also revealed that the process favored physisorption as it was quick, spontaneous, and exothermic. Future work will include exploring other activation methods and targeting pollutants that utilize the AC’s magnetic properties for water treatment applications.

## Figures and Tables

**Figure 1 molecules-28-01511-f001:**
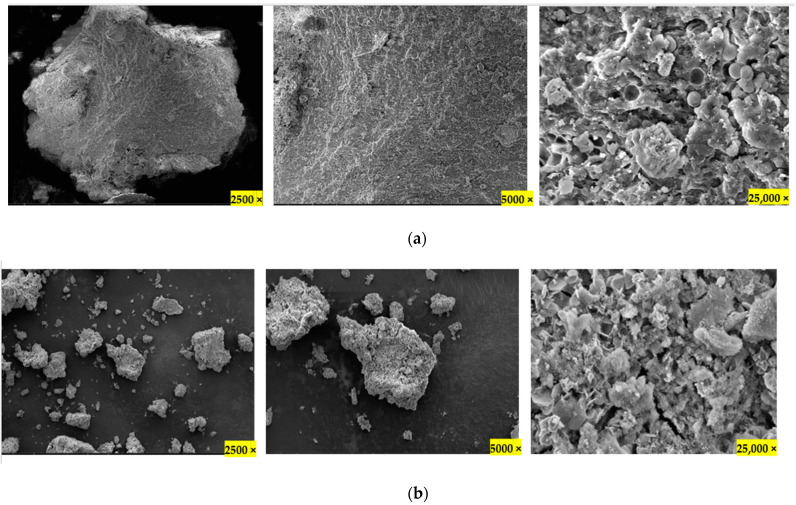
SEM images of Biosolids; (**a**) pristine (BS), (**b**) activated samples (KBS) in different magnifications.

**Figure 2 molecules-28-01511-f002:**
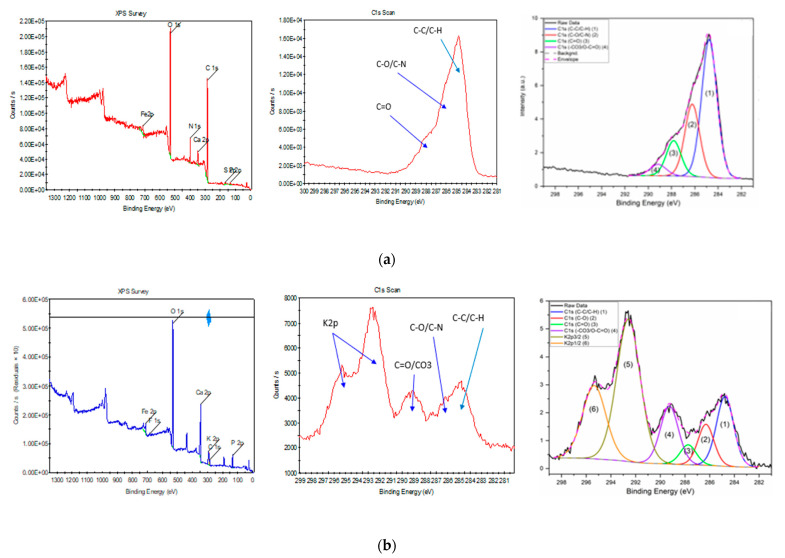
XPS results of Biosolids; (**a**) pristine (BS) and (**b**) activated samples (KBS).

**Figure 3 molecules-28-01511-f003:**
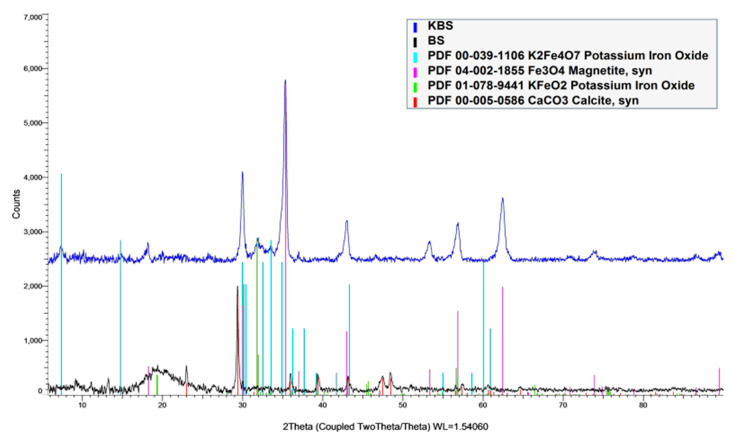
XRD results of of bothpristine (BS) and aactivated samples (KBS).

**Figure 4 molecules-28-01511-f004:**
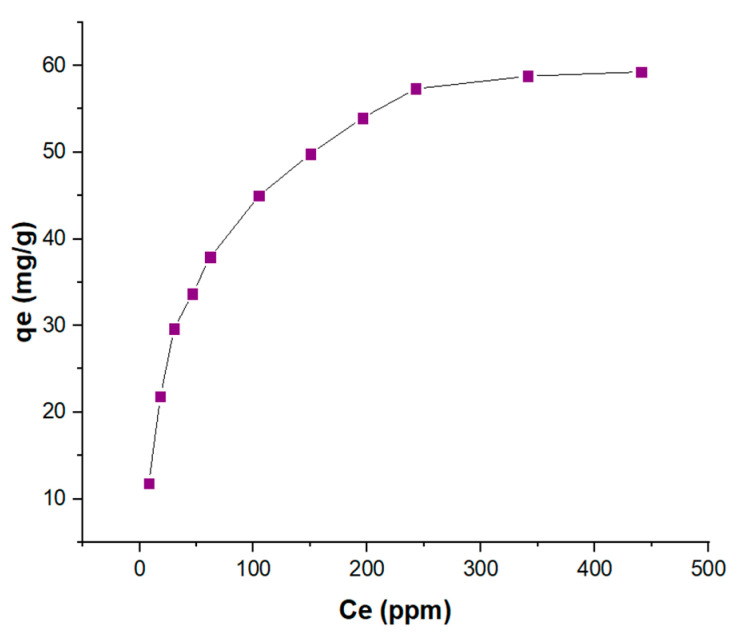
Maximum MB adsorption capacity (mg/g) of KBS at 40 °C and pH 7.

**Figure 5 molecules-28-01511-f005:**
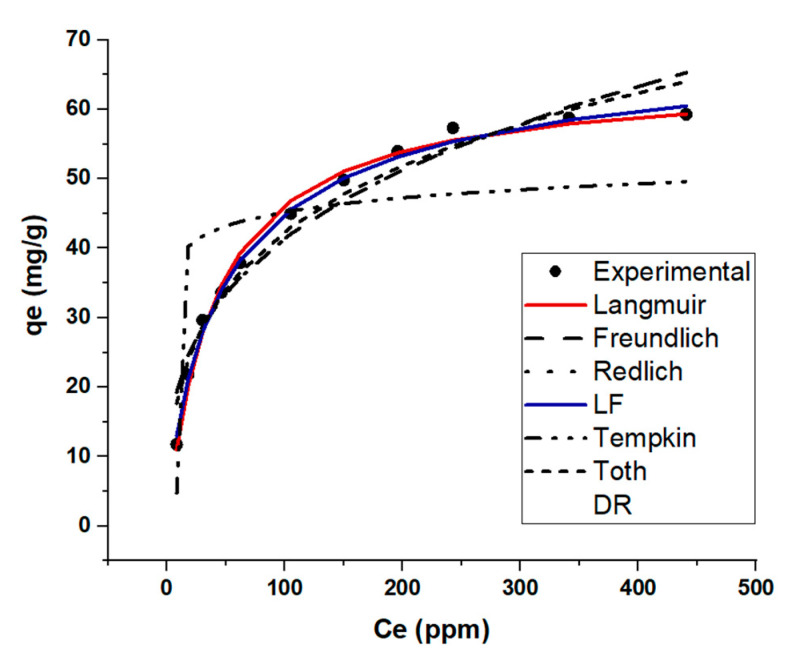
Isotherm modelling results of KBS: MB adsorption capacity at equilibrium (qe) Vs. final concentration.

**Figure 6 molecules-28-01511-f006:**
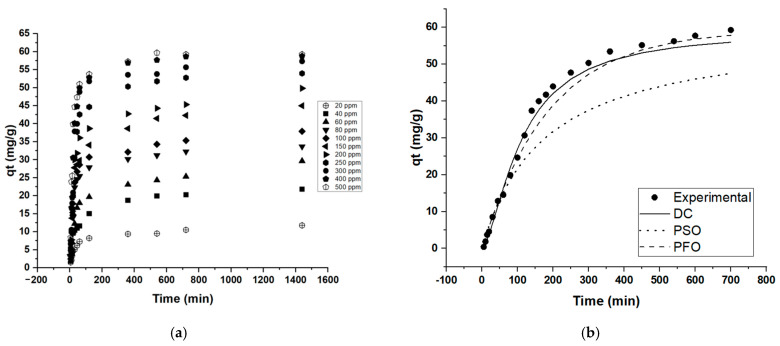
MB adsorption capacity (qt) Vs. time; (**a**) contact time trend, (**b**) best-fit kinetic models.

**Figure 7 molecules-28-01511-f007:**
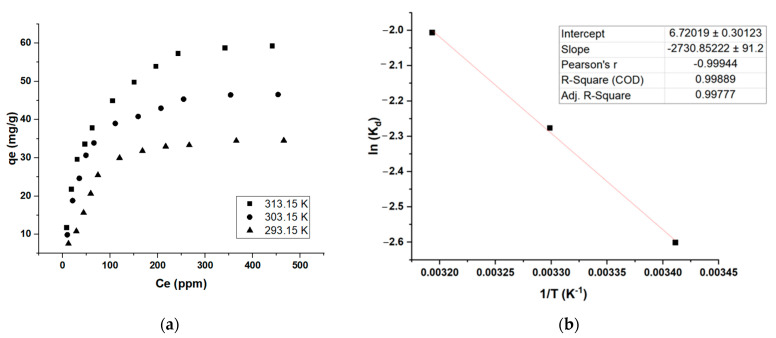
Thermodynamic calculations; (**a**) effect of temperature on MB adsorption, (**b**) Arrhenius plot showing ln (K_d_) Vs. 1/T using isotherm data generated at C_0_ = 500 ppm (maximum MB removal).

**Table 1 molecules-28-01511-t001:** Characteristics of biosolids before and after activation.

Sample	pH	Yield (%)	Conductivity (mS/cm)	Surface Charge (mV)	Surface Area (m^2^/g)	Pore Volume (cm^3^/g)	Pore Size (nm)
BS	8.13	-	0.0230 ± 0.002	−17.3 ± 0.20	0.010 ± 0.002	0.021 ± 0.01	2.12 ± 0.9
KBS	11.2 (un-adjusted)	33.2 ± 1.50	0.0760 ± 0.001	−20.3 ± 0.40	156.56 ± 25.5	0.235 ± 0.01	6.01 ± 1.4

**Table 2 molecules-28-01511-t002:** EDS microanalysis of elements on the surface of BS and KBS.

	Biosolid (BS)		Activated Biosolid (KBS)	
Element	Mass (%)	Atom (%)	Mass (%)	Atom [%]
C	40.34	53.05	4.2	9.42
N	5.6	6.31	0.94	1.81
O	33	32.58	28.35	47.74
Na	0.28	0.2	0.52	0.61
Mg	0.14	0.09	0.54	0.6
Al	0.16	0.09	0.78	0.78
P	1.66	0.85	7.11	6.18
S	1.58	0.78	0.06	0.05
Cl	0.12	0.05	0.2	0.15
K	0.23	0.09	4.76	3.28
Ca	10.1	3.98	20.66	13.89
Mn	0.27	0.08	0.62	0.3
Fe	6.53	1.85	31.06	14.98
Si	0	0	0.2	0.19

**Table 3 molecules-28-01511-t003:** XPS survey results on the surface of BS and KBS.

	BS	KBS
Name	Atomic %	
P 2p	1.21	9.36
C 1s	61.15	9.87
Ca 2p	1.64	13.6
O 1s	26.98	58.16
Fe 2p	1.16	3.42
K 2p	-	4.58
F 1s	-	1.01
N 1s	6.82	-
S 2p	1.04	-

**Table 4 molecules-28-01511-t004:** Adsorption capacities of waste adsorbents reported in literature.

Sample	Adsorption Capacity qe(mg/g)	References
Mixed municipal discarded material	7.2	[28]
Waste orange and lemon peels	38	[29]
Elaeagnus angustifolia seeds	72	[30]
Oil palm wastes	24	[31]
Coconut leaves	66	[32]
Ackee apple pod	49	[33]
BS	59.27	This study

**Table 5 molecules-28-01511-t005:** Best-fit isotherm models.

Isotherm Best-Fit Models	SSE	Parameters
LF	11.199	K_LF_: 2.830; n_LF_: 0.819; a_LF_: 0.039
Langmuir	18.461	K_L_: 1.615; a_L_: 0.0249

**Table 6 molecules-28-01511-t006:** Best-fit kinetic models.

Kinetic Best-Fit Models	SSE	Parameters
DF	0.161	qe: 59.27; K_DC_: 0.04; n: 1.547
PFO	0.188	qe: 59.27; K_1_: 0.005
PSO	0.311	qe: 59.27; K_2_: 9.71E-05

**Table 7 molecules-28-01511-t007:** Thermodynamic parameters for the adsorption of MB by KBS.

Temperature (K)	∆G ° (KJ/mol)	∆H° (KJ/mol)	∆S° (KJ/mol.k)
293.15	−2.243	−0.273	0.0067
303.15	−2.310		
313.15	−2.377		

## Data Availability

Research data not available in the manuscript can be obtained from the corresponding author through email.

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
