# Peer review of "Removal of Methylene Blue from Water Using Magnetic GTL-Derived Biosolids: Study of Adsorption Isotherms and Kinetic Models"

_molecules, 2023, doi:10.3390/molecules28031511_

Round 1

Reviewer 1 Report

Zuhara et al., reported the removal of methylene blue dye by utilizing a magnetitic gas to liquid-derived biosolids in their entitled research article “Removal of methylene blue from water using magnetic GTL derived biosolids: Study of adsorption isotherms and kinetic models. Though the manuscript is quite relevant and interesting I have suggestions for the authors:

1.     There is no continuity in the first two sentences in the abstract sections. Rephrase.

2.     In the abstract section, the authors have used the abbreviation “SEM-EDS, XPS, XRD, and BET” without introducing them first. Please introduce it.

Same with EDS and XPS

3.     In line 71, the objective must be in paragraphs, not bullets.

4.     Graphical abstract should not be in the main text.

5.     Orientation of the manuscript is not as per the journal.

6.     Fig, 1 SEM images are placed very casually. Requires improvement. 

Author Response

General comment: Zuhara et al., reported the removal of methylene blue dye by utilizing a magnetitic gas to liquid-derived biosolids in their entitled research article “Removal of methylene blue from water using magnetic GTL derived biosolids: Study of adsorption isotherms and kinetic models”. Though the manuscript is quite relevant and interesting I have suggestions for the authors:

Thank you for your kind remarks. Find below responses to your feedback.

Comment 1:  There is no continuity in the first two sentences in the abstract sections. Rephrase.

Response 1: Thank you for bringing this to our notice. It is now edited, please check Page No. 1, line number 11-14.

Comment 2: In the abstract section, the authors have used the abbreviation “SEM-EDS, XPS, XRD, and BET” without introducing them first. Please introduce it.

Response 2: Thank you for the recommendation. It is now edited, please check Page No. 1, line number 16-18.

Comment 3: In line 71, the objective must be in paragraphs, not bullets.

Response 3:  This is edited in the manuscript- Page no. 2, line 76-83.

Comment 4: Graphical abstract should not be in the main text.

Response 4: Thank you for the comment. The image has been moved before the text.

Comment 5: Orientation of the manuscript is not as per the journal.

Response 5: Thanks for the comment, this is considered, and changes were made in the manuscript.

Comment 6: Fig, 1 SEM images are placed very casually. Requires improvement. 

Response 6: Thanks for the recommendation. The figure is updated now in Page no. 4.

Reviewer 2 Report

In the present manuscript, authors have reported the removal of Methylene Blue from water by adsorption on activated biosolids obtained from GLT plant. The subject is interesting however there are certain quarries which must be satisfied before final publication:

 1) Authors have used many abbreviations which are either defined in later part of manuscript or not defined at all e.g LF is not defined and SSE is defined in later part of manuscript.  Authors must define an abbreviation where it is used for the first time and then strictly adhere to that for clarity in text.

2) In abstract, Lines 13-14: It seems that the abbreviation “KBS” is used for potassium carbonate but actually it is not the case. This makes the whole manuscript confusing. Define the abbreviation properly and clearly. Similarly define “BS” properly before its use in Table 1.

3) There are many typos/grammatical mistakes in the manuscript e.g Page 1 Line 16: The sentence “The characterization shows ….. upon activation.” needs to be revised. Place a full stop after “using a tesla meter.” instead of “–“ and start a new sentence after that.  This writing style is used throughout the manuscript. Authors should carefully revise the manuscript and unwanted use of “–“ should be avoided.

4) Last sentence of the abstract should be removed from the abstract and may be incorporated in the conclusion.

5) Page 1 Line 36: “the common methods include chemical oxidation….” Here an important method “micellar solubilization” is missing. Here are few relevant references:

i) Application of anionic-nonionic mixed micellar system for solubilization of methylene blue dye. J. Mol. Liq. 2022 https://doi.org/10.1016/j.molliq.2022.120958

ii) Mixed Micellar Solubilization of Naphthol Green B Followed by Its Removal from Synthetic Effluent by Micellar-Enhanced Ultrafiltration under Optimized Conditions Molecules 2022, 27, 6436

iii) Application of cationic-nonionic surfactant based nanostructured dye carriers: Mixed micellar solubilization.  J. Mol. Liq. 2021, 326 https://doi.org/10.1016/j.molliq.2021.115345

6)  Page 2 Lines 68-79 should be removed from the introduction.

7) Labeling (a) and (b) is missing in Figure 1.

8) Space between Fe and 2p in Table 3.

9) Table S2 appears before Table S1 in the text.

10) Section 2.3 is not clear. Authors should revise it carefully.

11) Table 5: KL, KLF, nLF, aLF and aL Should be written in accordance with equations.

12) In caption of Figure 6, is it KB or KBS?

13) Section 2.5 has many mistakes e.g. there is only one value of DH but in text it is written as DH values;  Ea is activation energy not Free energy etc. Authors should revise it carefully.

14) In Table 7 Units of DS are incorrect. Also J & K must be in capital letters.

15) Labeling (a) and (b) is missing in Figure 8.

16)  Page 10 line 350: The treated samples were washed and the pH was adjusted… hydrochloric acid. The samples were dried before further use.

17) Starting sentence (lines 419-420) of Section 3.4 should be revised.

18) Page 11 line 416: the mass of the…?.. in g.

19) Page 11 line 427: Adsorption at…… was analyzed….

20)  Page 12 line 434-435: adsorbed pollutant amount…..

21) UV-Visible spectra are not provided in the manuscript.

22) Why is study performed only on pH 7?

23) Conclusion and abstract are essentially similar. Rewrite the conclusions in accordance with the results obtained.

Author Response

General comment: In the present manuscript, authors have reported the removal of Methylene Blue from water by adsorption on activated biosolids obtained from GLT plant. The subject is interesting however there are certain quarries which must be satisfied before final publication:

Thank you for your kind remarks. Find below responses to your feedback.

Comment 1:    Authors have used many abbreviations which are either defined in later part of manuscript or not defined at all e.g LF is not defined and SSE is defined in later part of manuscript.  Authors must define an abbreviation where it is used for the first time and then strictly adhere to that for clarity in text.

Response 1:  The necessary changes are made based on the suggestions on Page 8, line 210 to 211. The abbreviations are also updated in the list.

Comment 2: In abstract, Lines 13-14: It seems that the abbreviation “KBS” is used for potassium carbonate but actually it is not the case. This makes the whole manuscript confusing. Define the abbreviation properly and clearly. Similarly define “BS” properly before its use in Table 1.

Response 2: This comment is thoughtful, thank you. Please check line 89 in Page no. 3 and in the abstract in Page no.1.

Comment 3: There are many typos/grammatical mistakes in the manuscript e.g Page 1 Line 16: The sentence “The characterization shows ….. upon activation.” needs to be revised. Place a full stop after “using a tesla meter.” instead of “–“ and start a new sentence after that.  This writing style is used throughout the manuscript. Authors should carefully revise the manuscript and unwanted use of “–“ should be avoided.

Response 3: The comments above are taken into account and changes can be seen in the abstract in Page no.1.

Comment 4: Last sentence of the abstract should be removed from the abstract and may be incorporated in the conclusion.

Response 4: This comment is addressed in the abstract in Page no.1.

Comment 5: Page 1 Line 36: “the common methods include chemical oxidation….” Here an important method “micellar solubilization” is missing. Here are few relevant reference

  1. i) Application of anionic-nonionic mixed micellar system for solubilization of methylene blue dye. J. Mol. Liq. 2022 https://doi.org/10.1016/j.molliq.2022.120958

  1. ii) Mixed Micellar Solubilization of Naphthol Green B Followed by Its Removal from Synthetic Effluent by Micellar-Enhanced Ultrafiltration under Optimized Conditions Molecules 2022, 27, 6436

iii) Application of cationic-nonionic surfactant based nanostructured dye carriers: Mixed micellar solubilization.  J. Mol. Liq. 2021, 326 https://doi.org/10.1016/j.molliq.2021.115345

Response 5: The above comment was taken into account and the necessary changes in lines 43-33.

Comment 6: Page 2 Lines 68-79 should be removed from the introduction.

Response 6: The comment is duly noted, and the bullets are removed, and the objectives are written in a paragraph in Page no.3, line 76 to 83.

Comment 7: Labeling (a) and (b) is missing in Figure 1.

Response 7: This comment is addressed in the edited Figure 1.

Comment 8: Space between Fe and 2p in Table 3.

Response 8: Necessary edit made in Table 3 according to your suggestion.

Comment 9: Table S2 appears before Table S1 in the text.

Response 9: Necessary change is made in the manuscript in section 2.3.

Comment 10: Section 2.3 is not clear. Authors should revise it carefully.

Response 10: Section 2.3 is heavily edited as per your suggestion.

Comment 11: Table 5: KL, KLF, nLF, aLF and aL Should be written in accordance with equations.

Response 11: The necessary changes are made in Table 5.

Comment 12: In caption of Figure 6, is it KB or KBS?

Response 12: The change is made in the caption.

Comment 13: Section 2.5 has many mistakes e.g. there is only one value of DH but in text it is written as DH values;  Ea is activation energy not Free energy etc. Authors should revise it carefully.

Response 13: All the necessary changes are made in Section 2.5.

Comment 14: In Table 7 Units of DS are incorrect. Also J & K must be in capital letters.

Response 14: All the necessary changes are made in Table 7 and intext in Section 2.5.

Comment 15: Labeling (a) and (b) is missing in Figure 8

Response 15: The addition is made in the figure 8.

Comment 16: Page 10 line 350: The treated samples were washed and the pH was adjusted… hydrochloric acid. The samples were dried before further use.

Response 16: The change is made in Page no. 11 in line 322.

Comment 17: Starting sentence (lines 419-420) of Section 3.4 should be revised.

Response 17: The section is rewritten in lines 406 to 410. 

Comment 18: Page 11 line 416: the mass of the…?.. in g.

Response 18: The change was made in Page no. 12, line 403.

Comment 19: Page 11 line 427: Adsorption at…… was analyzed….

Response 19: The change was made in line 419.

Comment 20: Page 12 line 434-435: adsorbed pollutant amount…..

Response 20: The change was made in line 429.

Comment 21: UV-Visible spectra are not provided in the manuscript.

Response 21: It is now provided in the supplementary file and a line has been added to section 3.3 has been added to the manuscript.

Comment 22: Why is study performed only on pH 7

Response 22: Thanks for this comment. The effect of pH is added as an objective and changes are made in the methodology, sections 3.3, line 392 to 393, results and discussion, section 2.2, line 172 to 177, and Figure S3 is added to the supplementary file showing the effect of pH on adsorption capacity.

Comment 23: Conclusion and abstract are essentially similar. Rewrite the conclusions in accordance with the results obtained.

Response 23: The conclusion and abstract sections are rewritten.

Reviewer 3 Report

Carbonaceous materials prepared from different precursors are widely used for the decontamination of dye-contaminated wastewater. The topic of this manuscript is interesting. However, major revisions are required and the comments are given below.

1.     Biomass derived biochars are widely applied in waste water treatment and other fileds. Many papers have been published on multifunctional biochars. More typical references are suggested to be cited for broad readers, e.g. Journal of Bioresources and Bioproducts 2022, 7 (2), 109-115; Journal of Bioresources and Bioproducts 2021, 6 (4), 292-322; Inorganic Chemistry Frontiers 2022, 9, 6108-6123; New Journal of Chemistry 2022, 46, 10844-10853.

2.     It would be better to give the full name of various technologies for SEM-EDS, XPS, XRD, and BET in the abstract.

3.     Please double check the whole manuscript to remove typos, such as “59.27 mg/g-” in line 21.

4.     “dye removal” or “waste water treatment” is suggested to be added as a keyword.

5.     The high-resolution C1 s spectra in Figure 3 are suggested to be deconvoluted into individual peaks, please refer and cite Biochar 2022, 4 (1), 50.

6.     Issues like pH and absorption time on the adsorption capacity are suggested to be studied in “2.2. Adsorption experiments”.

7.     Why potassium carbonate was added to the sample at an impregnation ratio of 1:3 (sample: activating agent)? How about the impregnation ratio on the specific surface area and adsorption capacity of KBS?

8.     How about the adsorption capacity of KBS compared to other carbonaceous adsorbents?

9.     Please pay attention to the writing of references, especially the writing of journal titles and page numbers.

Author Response

General comment: Carbonaceous materials prepared from different precursors are widely used for the decontamination of dye-contaminated wastewater. The topic of this manuscript is interesting. However, major revisions are required, and the comments are given below.

Thank you for your kind remarks. Find below responses to your feedback.

 Comment 1:    Biomass derived biochars are widely applied in wastewater treatment and other fileds. Many papers have been published on multifunctional biochars. More typical references are suggested to be cited for broad readers, e.g. Journal of Bioresources and Bioproducts 2022, 7 (2), 109-115; Journal of Bioresources and Bioproducts 2021, 6 (4), 292-322; Inorganic Chemistry Frontiers 2022, 9, 6108-6123; New Journal of Chemistry 2022, 46, 10844-10853. 

Response 1: This comment is taken into account and added to the paper in lines 55 to 56.

Comment 2: It would be better to give the full name of various technologies for SEM-EDS, XPS, XRD, and BET in the abstract.

Response 2: This comment is addressed in Page no. 1, line 16 to 18.

Comment 3: Please double check the whole manuscript to remove typos, such as “59.27 mg/g-” in line 21.

Response 3: This is edited, please check line 24.

Comment 4: “dye removal” or “waste water treatment” is suggested to be added as a keyword.

Response 4: Taking the suggestion into notice, new keywords ‘dye removal’ is added to the keyword list

Comment 5: The high-resolution C1 s spectra in Figure 3 are suggested to be deconvoluted into individual peaks, please refer and cite Biochar 2022, 4 (1), 50.

Response 5: Thanks for the comment. Deconvolution was carried out for both samples and the results are presented in section 2.1, line 115 to 129 and figure 3 is also changed accordingly. Since this section was added newly to the paper, a new author who is an expert in deconvolution, Yahya Zakaria is added to the paper.

Comment 6: Issues like pH and absorption time on the adsorption capacity are suggested to be studied in “2.2. Adsorption experiments”.

Response 6: The effect of pH is added as an objective and changes are made in the methodology, sections 3.3, line 392 to 393, results and discussion, section 2.2, line 172 to 177, and Figure S3 is added to the supplementary file showing the effect of pH on adsorption capacity.

Regarding adsorption time, section 2.4 and figure 7, already discussed the effect of contact time on adsorption capacity of KBS.  

Comment 7: Why potassium carbonate was added to the sample at an impregnation ratio of 1:3 (sample: activating agent)? How about the impregnation ratio on the specific surface area and adsorption capacity of KBS?

Response 7: Thanks for this comment. Preliminary tests showed that other impregnation ratios 1:1 and 1:2 did not produce activated carbons with sufficient surface areas to be used for pollutant removal from water. This was the reason why these results were not added to the manuscript initially. However, the data is now available in the supplementary material, Table S1 and a text is added to the line 86 to 87 explaining the same.

Comment 8: How about the adsorption capacity of KBS compared to other carbonaceous adsorbents?

Response 8: Thank you for the comment but Table 4 already compared the adsorption capacity of KBS with six other adsorbents.

Comment 9:   Please pay attention to the writing of references, especially the writing of journal titles and page numbers.

Response 9: This was reviewed, and necessary changes were made.

Thank you so much for the review, we hope the revisions are satisfactory.

Reviewer 4 Report

Water pollution is a serious problem, and various pollutants, among them organic dyes, have raised the respective attention. The present paper addresses this issue by studying application of a biosolid waste obtained from wastewater treatment. The material has been activated and the adsorption characteristics towards Methylene Blue determined. The results seem to show that the sorbent has comparable properties to many other recently presented sorbents obtained from waste materials.

The study has a few major problems. The writing and presentation is very careless, and the presented data contain several clear mistakes. With all respect, but in my opinion, corresponding author, professor McKay probably have not checked in detail the submitted manuscript.

Some of the problems

1. Page 7: a LF model is mentioned many times. However, authors never introduced it.

2. Figure 6: Such representation is never used in decent papers. Please plot the experimental data as discrete symbols, and the model curves as continuous lines.

3. Figure 7b: the same as above. In addition: Many of the model curves, as displayed , are erroneous. They cannot be real computer fitted models, as they dont show mathematically possible functions.

4. Figure 8a: the data, as presented here, are incorrect, because they dont lead to the results shown in Figure 8b. Data and their presentation should be checked.

Author Response

Comment 1: Page 7: a LF model is mentioned many times. However, authors never introduced it.

Response 1: This comment is noted and necessary changes are made in Page No.8, lines 210 – 220.

Comment 2: Figure 6: Such representation is never used in decent papers. Please plot the experimental data as discrete symbols, and the model curves as continuous lines.

Response 2: Thank you for this comment, we have edited the figure as per the suggestion. The experimental points are kept as symbols and the experimental values in different types of lines.

Comment 3: Figure 7b: the same as above. In addition: Many of the model curves, as displayed, are erroneous. They cannot be real computer fitted models, as they dont show mathematically possible functions.

Response 3: The figure is changed accordingly. The model curves have been rechecked based on the equations in Table S3 in the supplementary file and the necessary changes have been made in Page 8, lines 231 to 236 and figure 7, b.

Comment 4: Figure 8a: the data, as presented here, are incorrect, because they dont lead to the results shown in Figure 8b. Data and their presentation should be checked.

Response 4: Thank you for the comment. The figures, caption and related methodology is section 3.5 are changed for further clarification. Additionally, the thermodynamic calculations that led to Figure 8b is represented in Table S4 in the supplementary file.

Round 2

Reviewer 1 Report

The authors have improved the manuscript and can be accepted in its current form.

Author Response

Thank you for accepting the revised version of the manuscript.

Reviewer 2 Report

Authors have incorporated all the suggested changes. The manuscript may be accepted after following corrections:

1) Add comma after "micellar solubilization" in line 44

2) Make correction in line 484 "Amjad, S.; Shaukat, S.; Rahman, H. M. A.; Usman, M.; Farooqi, Z. H.; Nazar, M. F"

Author Response

Comment 1: Add comma after "micellar solubilization" in line 44

Response 1: This comment is noted and the comma was added to line 47.

Comment 2:  Make correction in line 484 "Amjad, S.; Shaukat, S.; Rahman, H. M. A.; Usman, M.; Farooqi, Z. H.; Nazar, M. F"

Response 2: This comment is noted and the 5th reference is changed accordingly.

Reviewer 3 Report

The manuscript has been revised according to the comments and suggest to be accepted.

Author Response

(The authors gave the same response as above.)

Reviewer 4 Report

In the revised version, authors some of the criticized points, however they left other major erroneous parts in the manuscript. The paper should undergo a major revision.

1. The new figure 6 plots adsorption isotherms in non-linear representation. However, the x axis of the isotherm is the initial dye concentration, and the non-linear model is defined with equilibrium dye concentration. Therefore, the modelling results are incorrect.

2. Figure 7, plot b shows the experimental data of the time dependent sorption capacity, and three mathematical kinetic functions, plotted as lines (continuous, dash, dash-dotted). It is clearly seen that the mentioned lines have multiple break points, and therefore they cannot be the true representations of the mentioned mathematical models, contrary to what authors claim.

3. In Figure S5, authors copy paste a figure from another publication, without proper explanation and referencing and note of copyright permission. This figure, as displayed, makes the false impression that this is the data belonging to the present study, while it is not the case.

Author Response

Comment 1: The new figure 6 plots adsorption isotherms in non-linear representation. However, the x axis of the isotherm is the initial dye concentration, and the non-linear model is defined with equilibrium dye concentration. Therefore, the modelling results are incorrect.

Response 1: The reviewer is correct, we have modified the x-axis legend in Figure 6 to read final equilibrium concentration.  The data values in the Figure are now correct.

Comment 2: Figure 7, plot b shows the experimental data of the time dependent sorption capacity, and three mathematical kinetic functions, plotted as lines (continuous, dash, dash-dotted). It is clearly seen that the mentioned lines have multiple break points, and therefore they cannot be the true representations of the mentioned mathematical models, contrary to what authors claim.

Response 2: This comment is noted and the graph has been changed in Figure 7, b.

Comment 3: In Figure S5, authors copy paste a figure from another publication, without proper explanation and referencing and note of copyright permission. This figure, as displayed, makes the false impression that this is the data belonging to the present study, while it is not the case.

Response 3: Thanks for the comment, further clarification is made in Lines 367 – 369 in the revised manuscript and the supplementary material has also been updated with the copyright information.